# Population coding of conditional probability distributions in dorsal premotor cortex

Joshua I. Glaser [1,2], Matthew G. Perich [3,4], Pavan Ramkumar[2], Lee E. Miller [2,3,4] & Konrad P. Kording [2,3,4,5,6,7]

Our bodies and the environment constrain our movements. For example, when our arm is fully outstretched, we cannot extend it further. More generally, the distribution of possible movements is conditioned on the state of our bodies in the environment, which is constantly changing. However, little is known about how the brain represents such distributions, and uses them in movement planning. Here, we record from dorsal premotor cortex (PMd) and primary motor cortex (M1) while monkeys reach to randomly placed targets. The hand's position within the workspace creates probability distributions of possible upcoming targets, which affect movement trajectories and latencies. PMd, but not M1, neurons have increased activity when the monkey's hand position makes it likely the upcoming movement will be in the neurons' preferred directions. Across the population, PMd activity represents probability distributions of individual upcoming reaches, which depend on rapidly changing information about the body's state in the environment.

[1] Interdepartmental Neuroscience Program, Northwestern University, 320 E Superior St, Morton 1-645, Chicago, IL 60611, USA. [2] Department of Physical Medicine and Rehabilitation, Northwestern University and Shirley Ryan Ability Lab, 710 N Lake Shore Drive # 1022, Chicago, IL 60611, USA. [3] Department of Physiology, Northwestern University, 303 E Chicago Ave, M211, Chicago, IL 60611, USA. [4] Department of Biomedical Engineering, Northwestern University, 2145 Sheridan Rd, Evanston, IL 60208, USA. [5] Department of Applied Mathematics, Northwestern University, 2145 Sheridan Rd, Evanston, IL 60208, USA. [6] Department of Neuroscience, University of Pennsylvania, 415 Curie Blvd, Philadelphia, PA 19104, USA. [7] Department of Bioengineering, University of Pennsylvania, 210S 33rd St, Suite 240 Skirkanich Hall, Philadelphia, PA 19104, USA. Correspondence and requests for materials should be addressed to J.I.G. (email: joshglaser88@gmail.com)

To plan movements, we must incorporate knowledge of the state of our bodies within the current environment. For example, if we are standing in front of a wall, we cannot walk forwards; if our arm is fully outstretched, extending it further is not possible. Considerations like these make some movements more likely than others, resulting in probability distributions over possible movements (Fig. 1a). To understand everyday movement planning, it is essential to understand how the brain represents these probability distributions.

Several studies have investigated whether the brain represents probabilities during movement planning[1–4]. In most, subjects needed to decide to move in one of a small number of directions, and the probabilities of those choices were manipulated[1–3]. These studies have shown that neurons in several brain areas have higher firing rates when there is a greater probability of an upcoming movement planned in those neurons' preferred directions (PDs). Recently, we began to study how the brain represents a continuous probability distribution rather than probabilities of discrete movements[4]. We displayed a point cloud representing an uncertain target location for movement. When we increased the uncertainty, there was a broader recruitment of dorsal premotor cortex (PMd) neurons, suggesting that PMd activity can reflect a distribution of possible movements.

Still, there is a large gap between these previous experiments and the real world, which contains dynamically changing conditional probability distributions, i.e., probabilities dependent on some background knowledge (here, the current state of the body in the environment). As the body moves, the probability distributions of possible upcoming movements change. If the brain

is to make use of these conditional probability distributions, it must rapidly compute updated probability distributions. Are these conditional probability distributions represented in the motor cortex, and if so where? How does the population of neurons function to represent these rapidly changing probabilities?

Here, we record from PMd and primary motor cortex (M1) while macaque monkeys reach to a series of targets that are chosen approximately randomly within the workspace. At all times, the position of the hand relative to the borders of the workspace dictates a conditional probability distribution of possible upcoming target locations. Behaviorally, the latencies and trajectories of the monkeys' movements are affected by this distribution, suggesting that they use this information during movement planning. Critically, neurons in PMd, but not M1, reflect these conditional probability distributions of upcoming movements prior to individual reaches, suggesting that such distributions are incorporated by the planning areas of motor cortex when coordinating movement.

## Results

**Experiment and behavior**. To study conditional probability distributions about upcoming movements, we recorded from three monkeys with electrode arrays chronically implanted in PMd and/or M1 while conducting a random-target reaching experiment. Monkey T had an array in PMd, monkey C had an array in M1, and monkey M had arrays in both areas. In the experiment (Fig. 1b), the monkeys reached sequentially to four targets, before receiving a reward. About 200 ms after the cursor

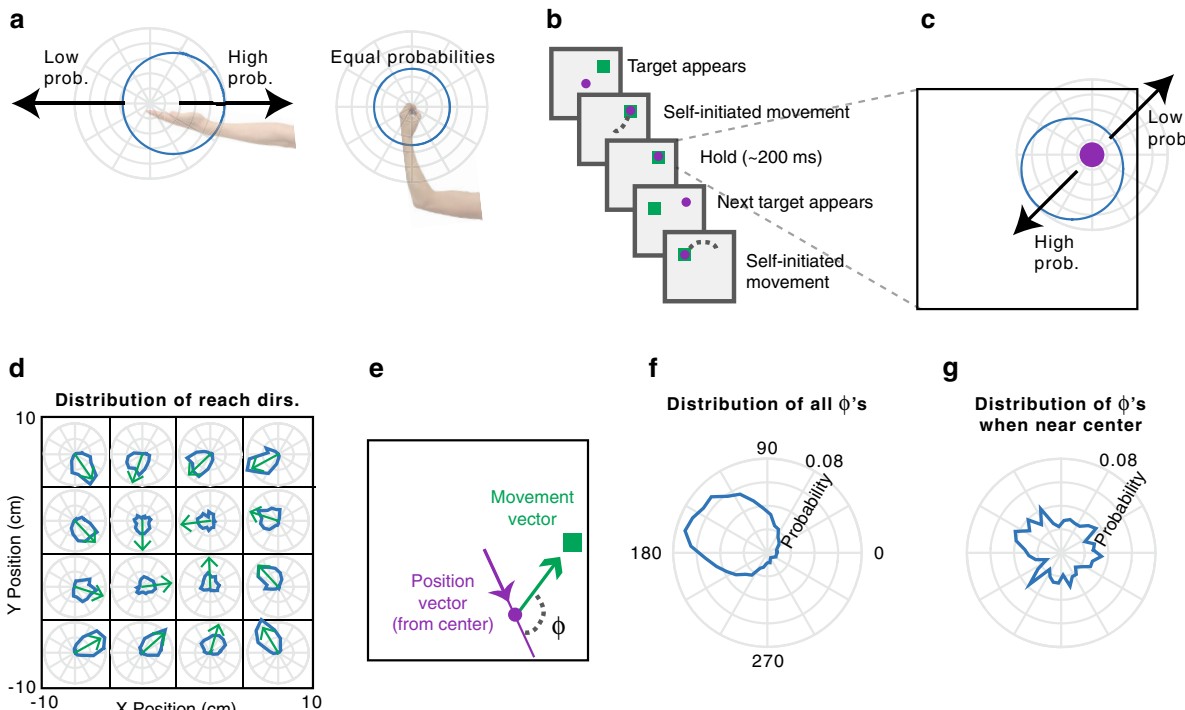

**Fig. 1** Experimental design and statistics. **a** If the arm is outstretched, the only possible arm movements are back toward the body (left). In other limb postures, it may be possible to move the arm in any direction (right). In blue, circular probability distributions are shown for the possible upcoming movements based on the current arm posture. **b** Experimental design. The monkey makes sequences of four reaches, briefly holding within each target box before the next target appears. **c** The current hand position limits the range of possible locations of the next target, due to the borders of the workspace and target presentation algorithm. **d** The probability distributions of upcoming reach directions (blue) from different areas of space (x and y divided into quartiles). Green arrows point toward the circular means of the distributions. **e** $\Phi$ is the angular difference between the upcoming movement vector (the vector that brings the hand to the target) and the current angular hand position (relative to the center of the workspace). **f** The probability distribution of $\Phi$'s from all hand positions. **g** The probability distribution of $\Phi$'s from initial hand positions within 2 cm of the center of the workspace

reached a given target, a new target appeared, to which the monkey could reach immediately. Due to the borders of the workspace, upcoming targets were more likely to be presented approximately opposite of the current hand position (Fig. 1c). That is, if a monkey's hand just landed on a target on the right side of the screen, it was more likely that the next target (and therefore, movement) would be to the left of this current hand position. Therefore, probability distributions in this experiment were conditioned on the hand's current position in the workspace at the time of target presentation.

The statistics of target presentation were not completely random within the workspace; rather, targets were slightly more likely to be selected in a clockwise direction (Fig. 1d; see Methods section). To summarize the dependence of upcoming target locations on the current hand position, we first found the angular position of the hand relative to the center of the workspace (Fig. 1e). We analyzed the distribution of $\Phi$'s: the angular differences between the current hand position vector and the upcoming movement vector (the vector that moves the hand to the target; Fig. 1e,f). A $\Phi$ of 180° signifies that the target was exactly opposite of the current angular hand position. Importantly, this distribution had a circular mean of 150° rather than 180° because of the slight clockwise bias in target selection. Additionally, we can see that the farther the hand position was from the center of the workspace, the more likely the upcoming target was to be in the opposite direction (Fig. 1d). When in a position near the center, there is little information about the upcoming target direction (Fig. 1g). We aimed to determine the effect of the conditional probability distributions of upcoming movements on behavior and neural activity.

To determine whether these conditional probability distributions influenced behavior, we first analyzed movement trajectories. The reaches generally did not go straight from one target to the next; they had some curvature that was influenced by the target probabilities (Fig. 2a,b). Early in the reach, trajectories were biased toward the expected target direction, defined as the most probable direction given the distribution of target presentations (i.e., 150° relative to the angular hand position; Fig. 1f). Additionally, the initial reach directions were more biased toward the expected target direction than simply toward the center of the workspace (Supplementary Fig. 1). Further, when the hand position was farther from the center (and the potential target distribution was more peaked) the magnitude of this bias was larger (Fig. 2b). This supports previous behavioral results showing that movement trajectories reflect uncertainty about the movement goal[5]. Our behavioral results suggest that the monkeys learned and accounted for the conditional probability distributions of possible upcoming targets when planning movements.

We then analyzed how the conditional target probabilities affected movement latencies. We found shorter latencies when the target appeared close to the vector of the expected direction (Fig. 2c; Monkey M, Pearson's $r = 0.26$, $p < 1e-10$; Monkey T, $r = 0.20$, $p < 1e-10$; Monkey C, $r = 0.041$, $p = 0.0045$). Note that this result is opposite of what we would expect due to momentum from the previous movement, as the expected direction is generally approximately opposite of the previous movement. Moreover, the distance of the hand from the center also affected the latency. For initial hand positions farther from the center (resulting in a tighter probability distribution), there was a larger latency difference between reaches to targets in expected and unexpected directions. (Fig. 2d; Monkey M, $p = 2.5e-5$; Monkey T, $p = 0.011$; Monkey C, $p = 0.068$). It is important to note that the latency and trajectory results are not independent. Since the latency is defined as the time to reach a velocity threshold (see Methods section), the monkeys could have shorter latencies when the initial trajectory was closer to the direction of the target, since

they didn't need to change direction. Overall, the monkeys' behaviors suggest that the motor system began movement preparation towards highly probable directions prior to target appearance.

**PMd neurons are modulated by upcoming movement probabilities.** Given that the conditional probability distributions about the potential upcoming movements affected behavior, we asked whether PMd represented this information in two monkeys. If PMd represents these probabilities, then we would expect neural activity preceding target onset to be modulated based on the anticipated possible target locations.

When observing peristimulus time histograms (PSTHs; Fig. 3a–c, Supplementary Fig. 2a-c for individual monkeys), we found some "potential-response" (PR) neurons (nomenclature as in ref. [6]). As expected for PMd neurons, these neurons' activity increased when a target was presented near their PDs (Fig. 3a). Crucially, PR neurons' activity was also modulated prior to target presentation by the range of possible upcoming movements. When the angular hand position was opposite these neurons' PDs (causing a higher probability that the upcoming target would be near the PD), pre-target activity increased. That is, for PR neurons, the red traces in Fig. 3b, c were elevated prior to target onset. Note that the activity prior to target onset in Fig. 3a for PR neurons is due to the correlation between the upcoming target and the current hand position; monkeys were apparently able to anticipate the upcoming reach direction. We also found "selected-response" (SR) neurons, whose activity was significantly modulated only after target presentation. That is, for SR neurons, the red and blue traces barely differed prior to target onset (see Supplementary Fig. 3 for an explanation of the slight difference before target onset in the PSTHs). Thus, PSTHs suggest that a subset of PMd neurons is modulated by the probability of upcoming movements, which seems to form part of the monkey's movement planning.

To analyze the factors contributing to neural activity more rigorously, we used a generalized linear modeling (GLM) approach. This approach can inform us whether the current hand position (and consequent probability distribution of upcoming target locations) significantly modulated neural activity above potential confounds related directly to the previous and upcoming movements. The GLM found that 13% (99/770) of neurons were PR and 42% (322/770) were SR neurons using this conservative classification approach (see Methods section for classification criteria, and Supplementary Fig. 4 for percentages with a less conservative criteria). For both types of neurons, the upcoming movement covariate began to matter after target onset (green trace in Fig. 3d, Supplementary Fig. 2d for individual monkeys). For PR neurons, but not SR neurons, the importance of the hand position covariate (purple trace) began to increase more than 200 ms prior to target onset, until target onset. The GLM analysis thus supports our PSTH results; prior to target presentation, PR neurons' activities are modulated by hand position, which determines the distribution of potential upcoming targets.

**The PMd population jointly represents the distribution of reaches.** An important question is how the neural population represents the probabilities about upcoming movements. We showed above that when the monkey's hand is in a position that makes an upcoming target more likely to appear near a PR neuron's PD, the neuron will have greater activity prior to target onset than it otherwise would (Fig. 3). This could be because the neural population activity is related to the statistical distribution of possible upcoming movements, conditioned on the current

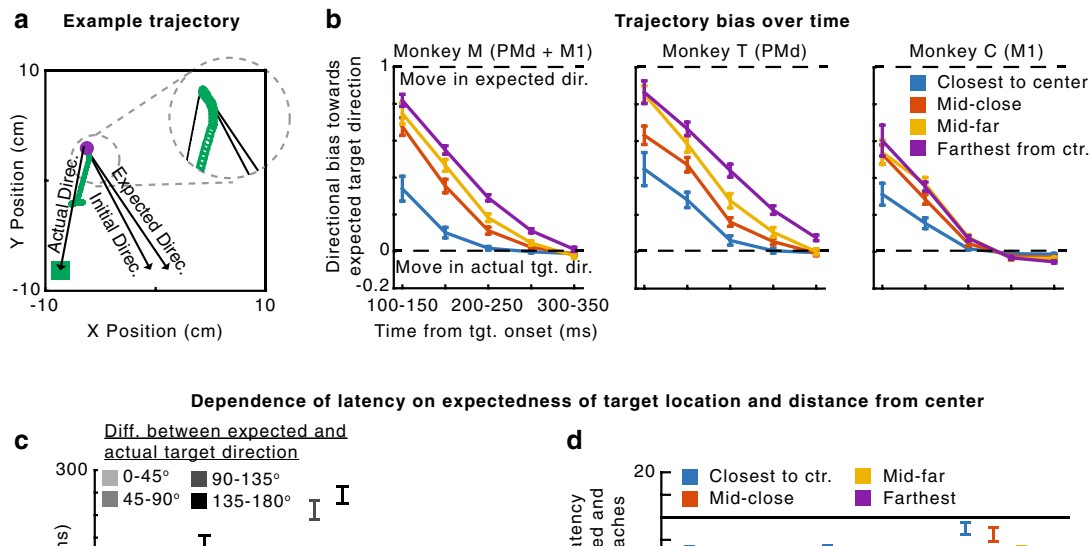

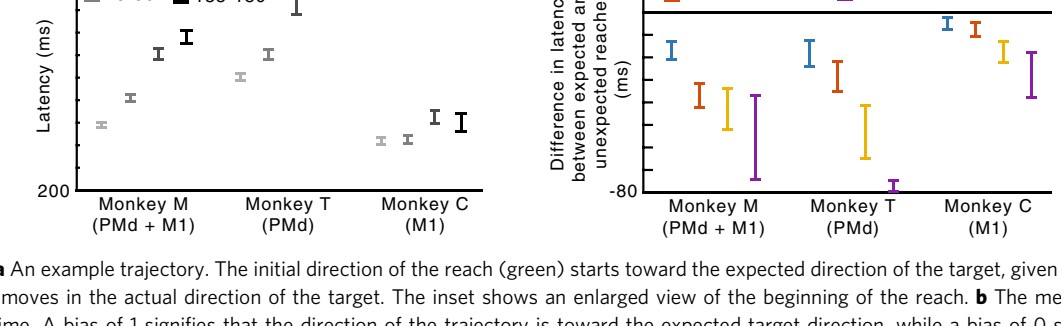

**Fig. 2** Behavior. **a** An example trajectory. The initial direction of the reach (green) starts toward the expected direction of the target, given the current hand position. It later moves in the actual direction of the target. The inset shows an enlarged view of the beginning of the reach. **b** The median bias of the trajectory over time. A bias of 1 signifies that the direction of the trajectory is toward the expected target direction, while a bias of 0 signifies that the direction of the trajectory is toward the actual target direction. Negative biases signify movement away from the expected direction. Different traces are shown for hand positions at varying distances from the center of the workspace. Error bars are standard errors of the median. **c** The mean latency of reaches as a function of the angular difference between the actual and expected target directions. **d** The difference in mean latency between expected and unexpected reaches (expected minus unexpected), depending on the hand's distance from the center. "Expected" reaches are those that had an angular difference between the actual and expected target directions of less than 60°. "Unexpected" reaches had an angular difference of more than 120°. In panels **c** and **d**, error bars represent SEMs. In panels **b** and **d**, distances from the center are divided as follows: "closest" is 0–20% of distances from the center, "mid-close" is 20–40%, "mid-far" is 40–60%, and "farthest" is 60–100%. We used these divisions for plotting, rather than standard quartiles, to ensure that there were "unexpected" reaches in each bin. When using standard quartiles, there were no "unexpected" reaches for some monkeys in the last quartile (greatest 25% of distances from the center) because when very far from the center, the next target cannot be in an unexpected direction (farther away from the center)

state. Alternatively, the neural population could be using some type of heuristic to determine the likely location of the next target (e.g., assuming the next target will always be toward the center).

To understand how the neural population activity relates to the conditional probability distributions of possible movements, we calculated the average activity (across PR neurons and reaches) as a function of the current angular hand position relative to each neuron's PD (Fig. 4a, Supplementary Fig. 5 for individual monkeys). Neural activity during the 100 ms prior to target onset closely reflected the statistics of possible target locations (Fig. 4b). The peak angle of the neural activity was not significantly different than 150°, the most likely Φ determined by the experimental design. This finding was not simply due to the correlation between the previous and upcoming movements or the correlation between the hand positions and upcoming movements (Supplementary Fig. 5). Moreover, when only looking at reaches starting near the center, activity prior to target-onset was clearly diminished (Fig. 4c), reflecting the lower and more uniform probabilities of upcoming reaches (Fig. 4d). Thus, when averaging across reaches and neurons, the population does represent the distribution of upcoming reaches.

How do neurons function together to create this distribution of upcoming reaches? It is possible that individual neurons reflect this distribution, and thus the population does as well.

In this scenario, the firing rate of each neuron as a function of position would correspond to the probability of movement into its PD. Alternatively, the distribution could be created only by many neurons working in concert. In this scenario, not all individual neurons' activities would correspond to the probabilities of upcoming movements into their PDs, but activity across the population would represent upcoming movement probabilities.

To differentiate between these possibilities, we analyzed how neurons' activities as a function of position related to the neurons' PDs. When we look at neurons with an upward preferred direction, we see that many individual PR neurons do not have maximal firing rates at hand positions corresponding to a maximum probability of moving upwards (Fig. 4e). Rather, these neurons have different "preferred positions", spanning many different areas at which upward movements are possible. However, when the activity of all these neurons is summed, the activity as a function of position closely matches ($r = .94$) the probability of an upward movement as a function of position (Fig. 4f). This suggests the movement probabilities are represented across the population rather than by individual neurons. Conversely, we can look at reach PDs relative to preferred angular position (the angular hand position leading to peak activity). When we orient the preferred position to be down, we see that

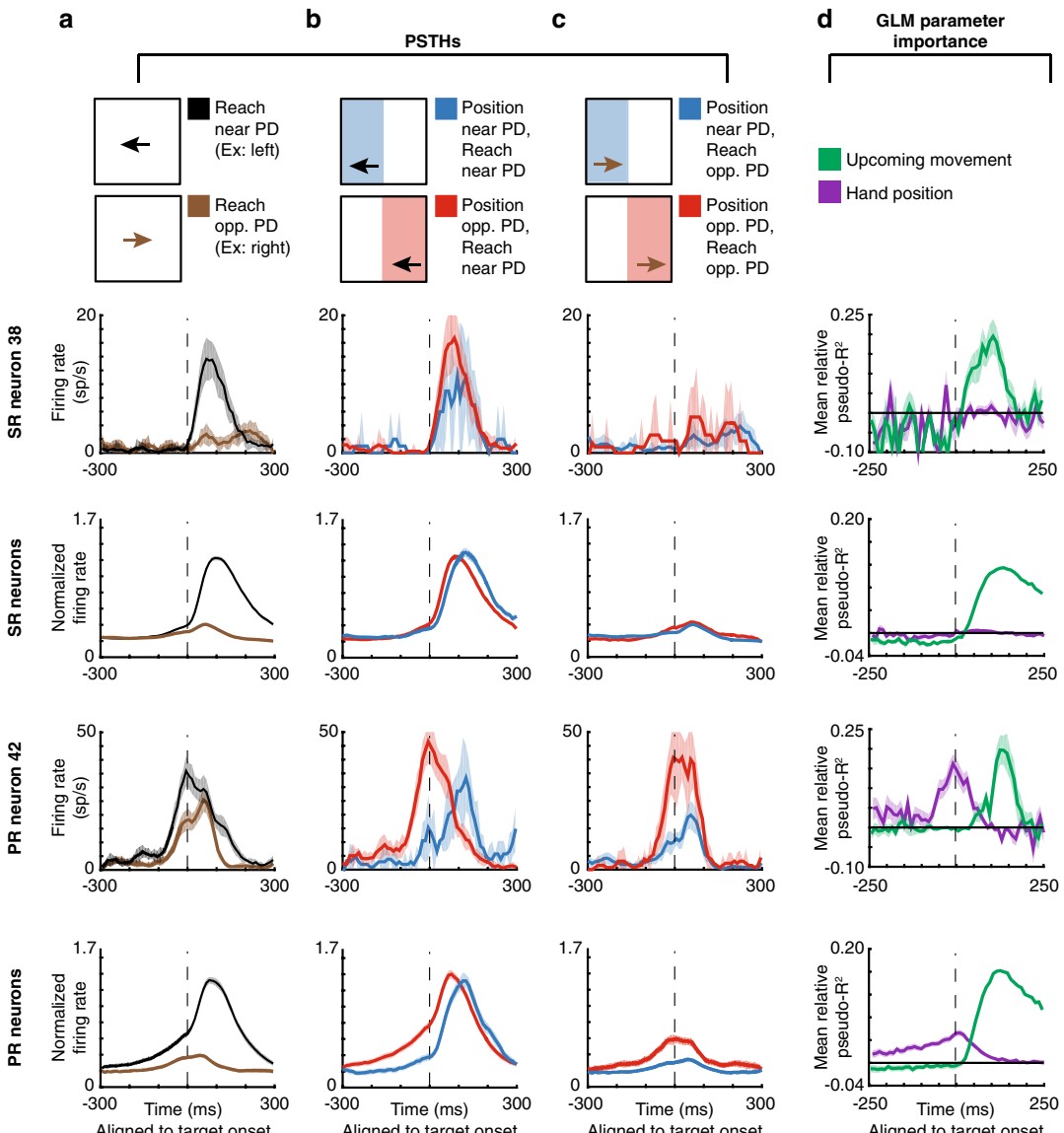

**Fig. 3** PMd PSTHs and GLM results. First row: A selected-response (SR) neuron. Second row: Normalized averages of SR neurons. Third row: A potential-response (PR) neuron. Bottom row: Normalized averages of PR neurons. **a–c** Peristimulus time histograms (PSTHs) for PMd neurons, aligned to target onset. Shaded areas represent SEMs. **a** PSTHs of reaches near the preferred direction (PD; black) vs. opposite the PD (brown). **b** PSTHs of reaches near the PD, with a starting hand position near the PD (lower probability of moving near the PD; blue) vs. a position opposite the PD (higher probability of moving near the PD; red). **c** PSTHs of reaches opposite the PD, with a starting hand position near the PD (blue) vs. a position opposite the PD (red). **d** We utilized a generalized linear model (GLM) to control for confounds in the PSTHs, including different distributions of starting positions, upcoming movements, and previous movements. Here, we show the importance of parameters in the GLM, across time, for PMd neurons. We show mean relative pseudo-$R^2$ over time, of the upcoming movement (green) and hand position (purple) covariates. For the 2nd and bottom row, shaded areas represent SEMs across neurons. For individual neurons, shaded areas represent the standard deviation across bootstraps

there are a wide range of reach PDs, reflecting possible upcoming movements from an initial downward position (Fig. 4g). The PDs of neurons are distributed approximately in proportion to how likely upcoming movement directions are (Fig. 4h). The population of PR neurons in PMd works together to represent the probability distribution of available upcoming movements given the current hand position.

**Single-reach representation of probability distributions.** While we have shown that the PMd population represents conditional probability distributions averaged across trials, we also want to know what is occurring prior to single reaches. Because we

recorded many neurons simultaneously, we can decode the monkey's intended movement prior to each reach. To do this, we first trained a naïve Bayes decoder to predict the reach direction (see Methods section) during the time period 50–200 ms after target presentation. We then used this decoder (with firing rate rescaling due to differing firing rates before and after target presentation; see Methods section) to estimate what movement the neural population was planning in the 100 ms prior to target presentation. Note that this method assumes that the PDs of neurons stayed the same between these two time periods. While PMd neurons are known to have different PDs during preparation and movement[7–9], both our time periods were during

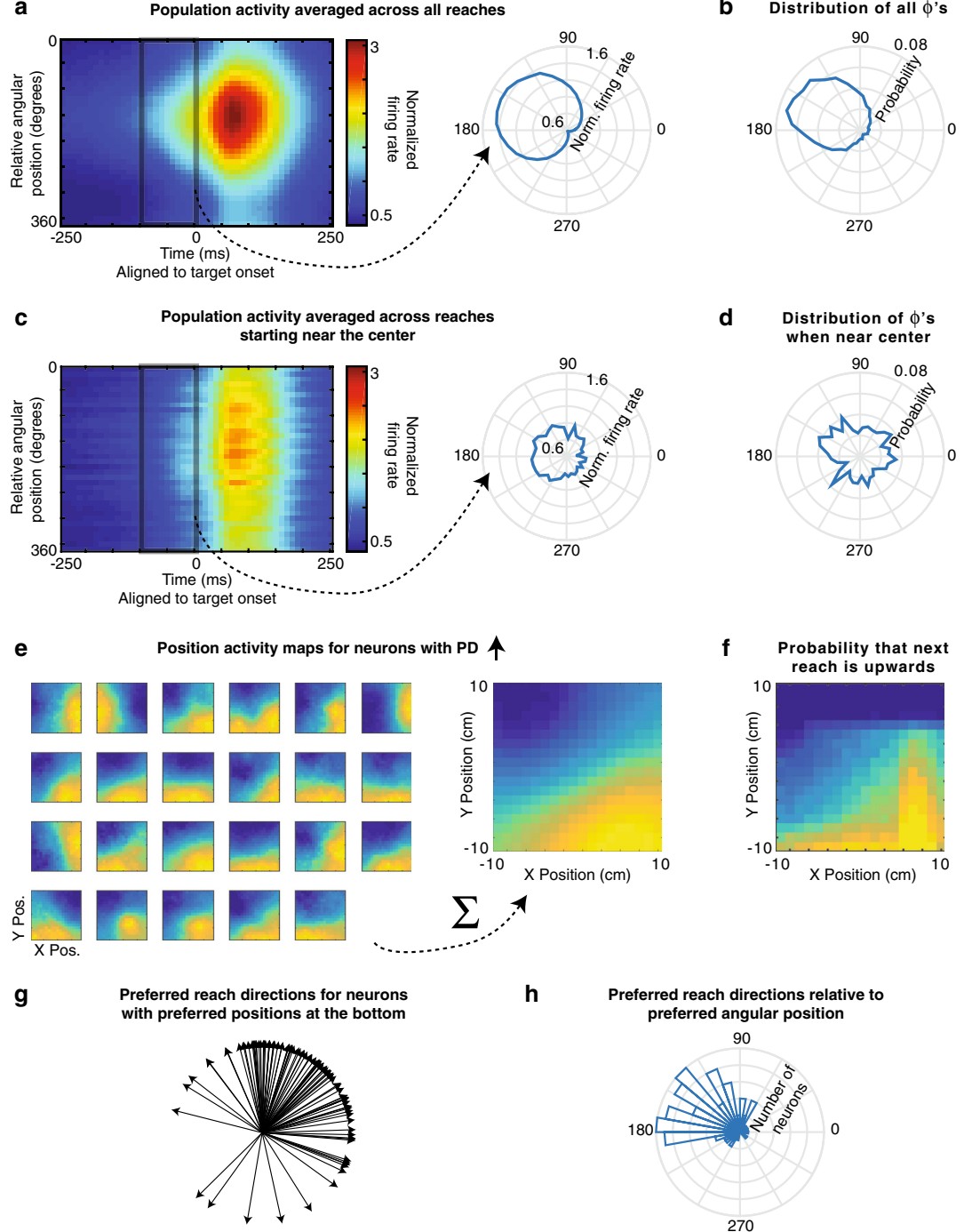

**Fig. 4** The PMd population jointly represents the distribution of upcoming reaches. **a** Left: The average normalized firing rate of all PR neurons, over time, as a function of relative angular hand position. For each neuron, the relative angular position is the preferred direction of the neuron minus the angular hand position. Activity is normalized and averaged across all PR neurons. Right: The average normalized firing rate in the 100 ms prior to target onset, plotted as a function of the relative angular hand position. **b** The distribution of upcoming movement directions relative to angular hand positions. This is duplicated from Fig. 1f for easy comparison with panel **a**. **c**, **d** Same as panels **a** and **b**, but for only for reaches starting near (within 2 cm of) the center. Note that panel **d** is duplicated from Fig. 1g. **e** Left: Position activity maps for example PR neurons with preferred movement directions that are oriented upwards. The position activity maps show the neurons' activity as a function of hand position (blue is low; yellow is high) from −100 to 50 ms surrounding target onset. Right: The sum of the position maps for all PR neurons, when their preferred directions are oriented upwards. **f** A map showing the probability that the next movement will be upwards, as a function of initial hand position. **g** Preferred reach directions for all PR neurons, when space is rotated so that their preferred hand angular position is oriented to be at the bottom (270°). **h** A histogram of preferred reach directions of all PR neurons relative to their preferred angular hand position (the reach PD minus the preferred angular hand position)

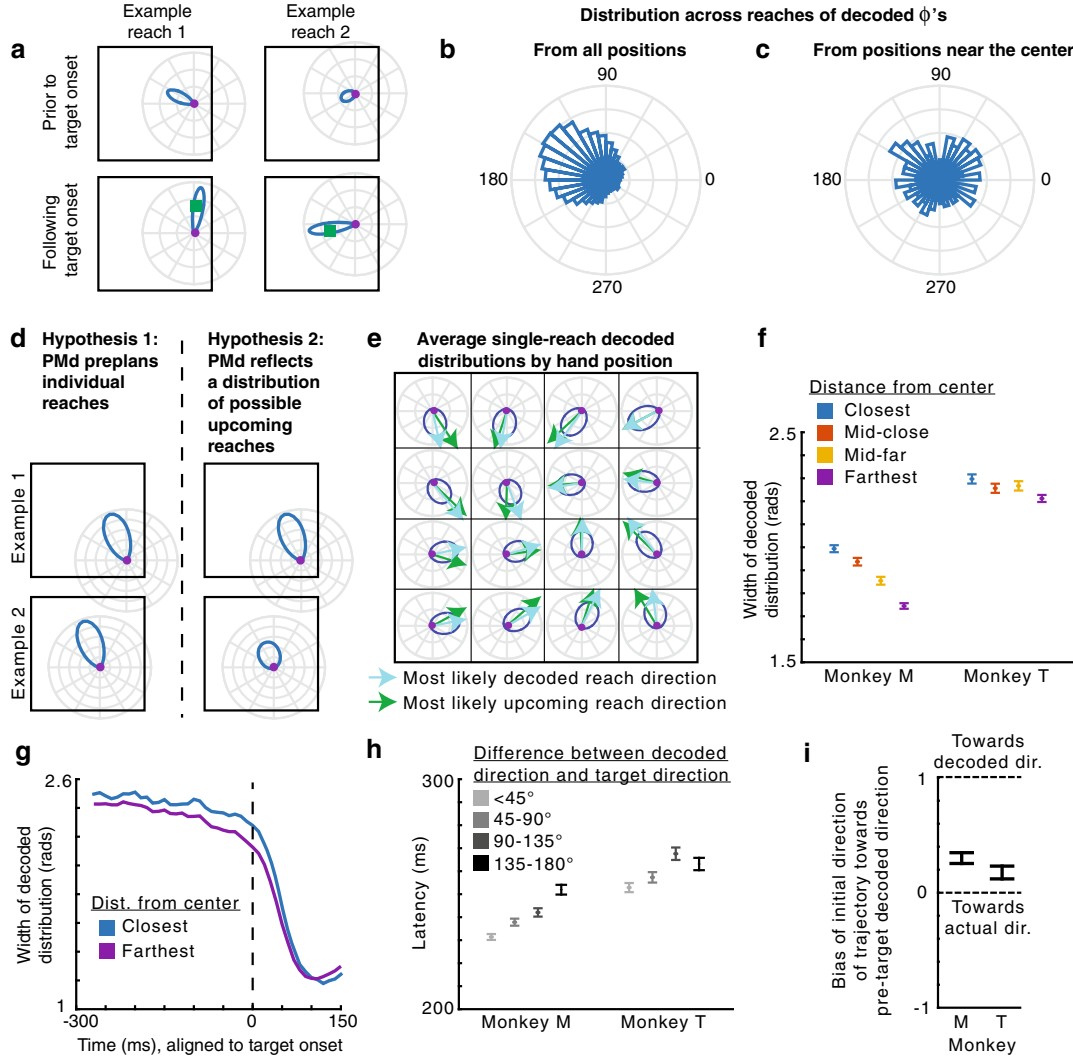

**Fig. 5** PMd population activity represents the distribution of upcoming movements—single reach decoding. **a** The distribution of decoded reach directions (blue) from the population of PMd neurons for two example reaches (left and right), before and after target onset (top and bottom). The purple circle is the current hand position, and the green square is the target location. **b** The distribution of decoded $\Phi$'s across all reaches. The decoded $\Phi$ is the pre-target decoded reach direction relative to the hand's angular position (same as $\Phi$ from Fig. 1, except with decoded direction instead of actual reach direction). **c** The distribution of decoded $\Phi$'s across reaches starting within 2 cm of the center. **d** The predictions of two hypotheses (left and right), shown for two different hand positions (example 1 vs. example 2). **e** Average pre-target decoded reach direction distributions as a function of hand position. The displayed distributions have the average width and peak angle of all decoded distributions from hand positions within the grid square. Light blue arrows point toward the circular means of these distributions. Green arrows point toward the most likely reach direction (from Fig. 1d). **f** The full width at half maximum of the pre-target decoded distribution as a function of hand distance from the center of the workspace. Distances from the center are binned as in Fig. 2. Error bars are SEMs. **g** Width of the decoded distributions over time, for starting positions that are closest (blue) and farthest (purple) from the center. **h** Latency of the reach as a function of the angular difference between the pre-target decoded direction and the actual target direction. Error bars are SEMs. **i** The bias of the initial trajectory of the reach (100–150 ms from target onset) toward the pre-target decoded direction. A bias of 1 signifies the initial trajectory is toward the decoded direction, while a bias of 0 is toward the actual target direction. Negative values are away from the decoded direction. 95% confidence intervals, computed via bootstrapping, are shown

preparation. Thus, we believe it is reasonable to use knowledge about neurons' PDs after target onset to decode planning prior to target onset.

As expected, the planned reaches decoded prior to target onset were usually approximately opposite of the current angular hand position (Fig. 5a,b). This can be seen in example trials (Fig. 5a), where the pre-target decoded reach direction was to the left when the hand position was on the right, while the post-target decoded reach direction was toward the target. Moreover, the distribution (across reaches) of pre-target decoded reach directions relative to the angular hand position approximately represented the experimentally-defined distribution of target presentations

determined by the current hand position (Fig. 5b,c; compare to Fig. 1f,g).

There are two explanations for PMd's apparent representation of the distribution of potential upcoming movement directions (Figs. 4a and 5b). One hypothesis is that PMd preplans one particular reach prior to a given target presentation, with a consistent level of confidence across reaches. That is, the PMd population does not actually represent a distribution of reaches on single trials, but averaging across trials yields the observed distribution. The alternative hypothesis is that PMd represents a probability distribution of possible movements prior to single reaches.

To distinguish between these two hypotheses, we looked at how single reach decoding depended on potential upcoming movements. The output of our probabilistic decoding method is a probability distribution reflecting the animal's movement intention encoded by the neural population for single reaches. Uncertainty in the population about the upcoming movement will make this distribution wider. We used the width of the decoded distribution to distinguish between the two hypotheses. If a single movement is being planned, with a consistent level of (un)certainty from reach to reach, the width of the decoded distributions should be approximately the same for every reach (Fig. 5d; Hypothesis 1). However, if PMd represents a distribution

of possible movements, when there are fewer possibilities for upcoming target locations (e.g., when the hand position is farther from the center), the distribution should be narrower than when there are more movement possibilities (Fig. 5d; Hypothesis 2). Indeed, the data show that the decoded distributions prior to target onset are narrower when the hand position is farther from the center (Fig. 5e–g; Monkey M, $p < 1e\text{-}10$; Monkey T, $p = 1.8e\text{-}6$). Thus, the decoded distributions suggest that PMd does represent a distribution of possible movements prior to single reaches.

Our decoding results provide insight into how neural activity prior to target onset influences individual upcoming reaches. The decoded pre-target reach directions were predictive of the monkeys' subsequent behavior. First, when the decoded direction was closer to the true target direction, reach latencies were shorter (Fig. 5h; Monkey M, $p < 1e\text{-}10$; Monkey T, $p = 2.1e\text{-}5$). Second, the initial direction of many reaches was initially biased toward the pre-target decoded reach direction (Fig. 5i; both monkeys $p < 0.05$ using bootstrapping). Additionally, the uncertainty of the decoded distributions influenced upcoming reaches. For targets in an expected upcoming direction, latencies were shorter when the width of the decoded distribution was narrower (Monkey M, $p < 1e\text{-}10$; Monkey T, $p = 3.3e\text{-}4$). These decoding results could provide a neural explanation for our observed latency and trajectory behavioral effects (Fig. 2). Overall, our decoding results provide insight into the expectations represented by the neural population prior to target onset.

**Control: visuomotor rotation**. In our task, the probabilities of upcoming target locations were determined by the current hand position. Thus, it is theoretically possible that the neural activity could only be modulated by position for some purpose other than representing upcoming movements[10,11]. As a control, we used data where the monkeys performed a visuomotor rotation (VR) learning task, which changed the probabilities of upcoming movements for the same cursor position in the workspace. In this task, cursor feedback on the screen was rotated by 30° counter-clockwise relative to the hand movement. That is, if a target were directly upwards on the screen, the monkey would now need to

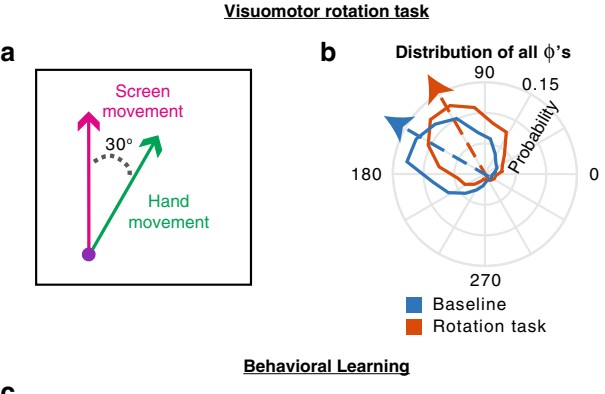

**Visuomotor rotation task**

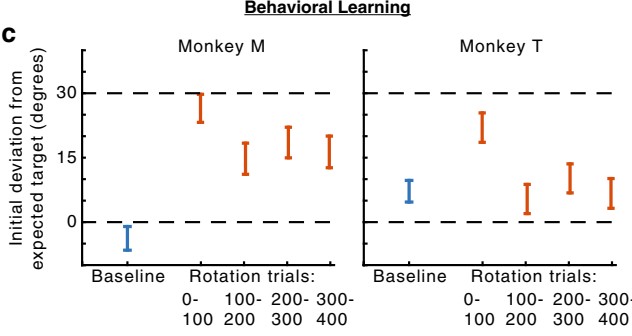

**Behavioral Learning**

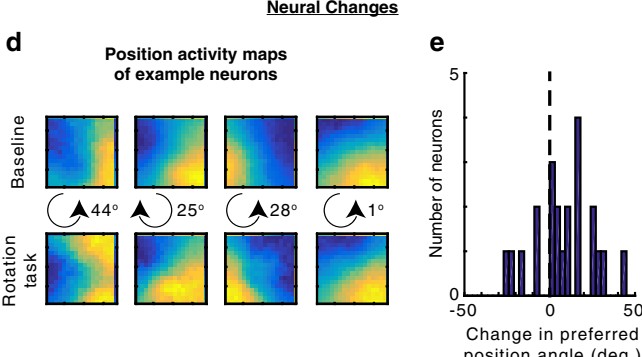

**Neural Changes**

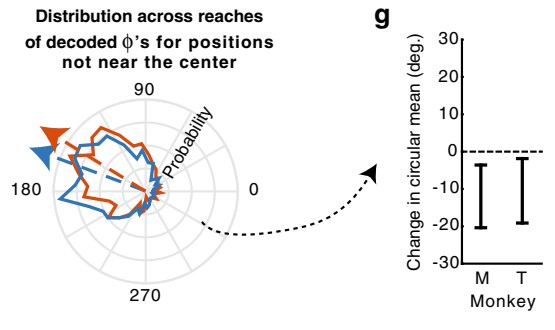

**Fig. 6** Visuomotor rotation control task. **a** The visuomotor rotation (VR) task. Movements on the screen (in the workspace) are rotated 30° counterclockwise relative to the hand movement. **b** The distribution of hand movements relative to the angular hand position in the workspace, i.e., $\Phi$'s, for the baseline task (blue) and the VR task (orange). Arrows point toward the circular means of the distributions. **c** The difference between the initial reach direction (100–150 ms from target onset) and the expected movement direction during the baseline task (blue) and different periods during the VR task (orange). The expected movement direction was the most likely upcoming movement direction given the current workspace position and movement statistics (panel **b**). Positive values mean the initial reach direction was counterclockwise of the expected target direction, meaning the monkey had not adapted. Error bars represent the circular SEM. **d** Position activity maps (activity as a function of position in the workspace) of example PR neurons in the baseline (top) and VR task in the second 2/3 of trials (bottom), as in Fig. 4c. In the middle, we show the direction (clockwise or counterclockwise) and magnitude of change of the preferred angular position. **e** The change in preferred angular position of all PR neurons (VR minus baseline). Positive means a counterclockwise shift. **f** The distribution of pre-target decoded reach directions relative to the hand's angular position (decoded $\Phi$'s) for positions not near the center (greater than the median distance). Decoding from the VR task used the second 2/3 of trials. **g** The difference between the circular mean of the distributions of decoded $\Phi$'s in panel **f**, between the baseline and VR tasks (VR minus baseline). Error bars represent 95% confidence intervals from bootstrapping

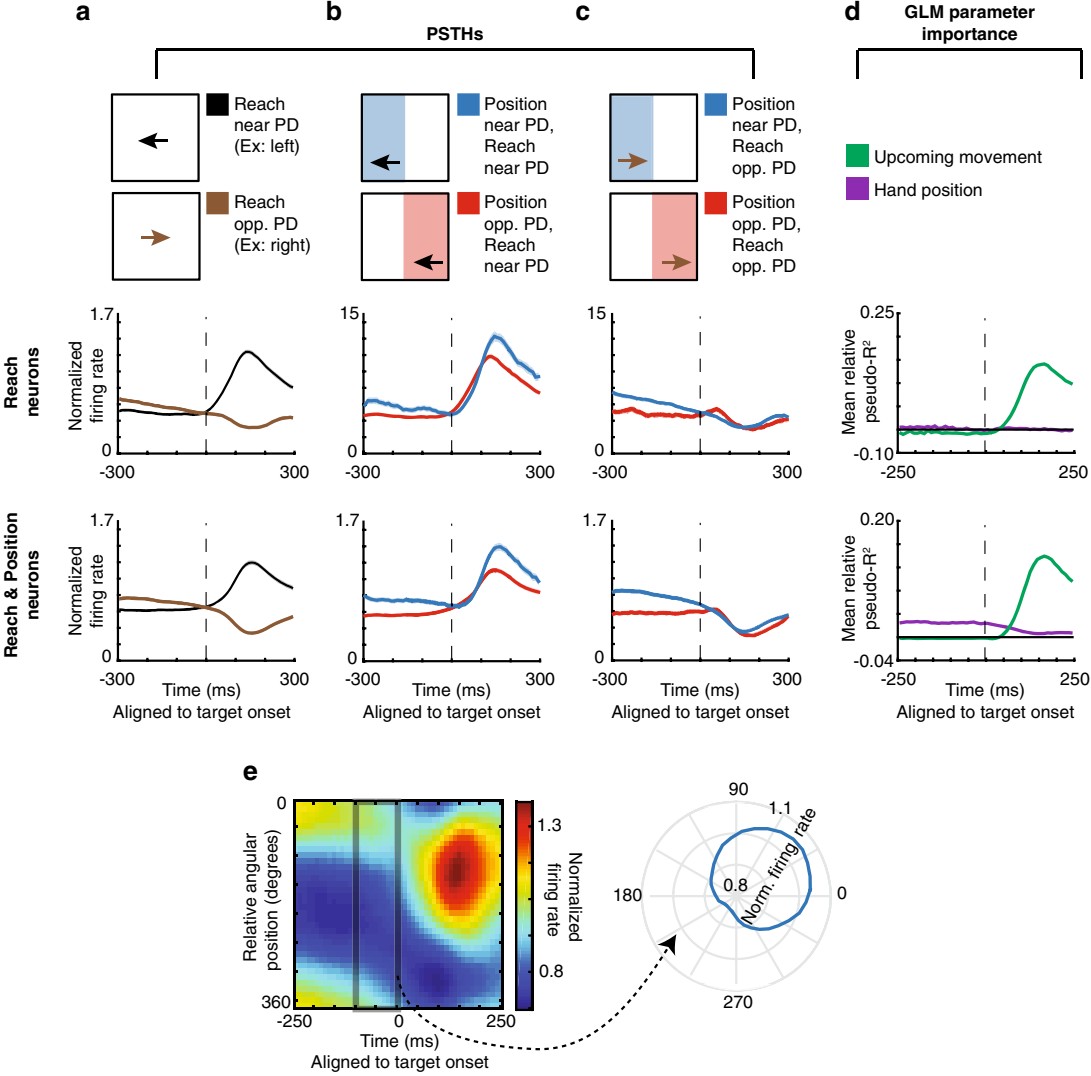

**Fig. 7** M1 does not reflect the probability of upcoming movements. **a–d** PSTHs and GLM results for M1 neurons. Columns have the same schematics as Fig. 3. First row of PSTHs: Normalized averages of reach neurons, defined as those neurons significant for movement during the late period, but not position in the early period of the GLM. This was the same criteria as for SR neurons in PMd. Second row: Normalized averages of reach & position neurons, defined as those neurons significant for movement during the late period, and position in the early period of the GLM. This was the same criteria as for PR neurons in PMd. Note that we did not use the same "SR/PR" nomenclature as PMd, because there was no evidence in the PSTHs of M1 neurons that position was used to represent potential upcoming movements. **e** Same schematic as Fig. 4a, but for M1 reach & position neurons. Left: The normalized average firing rate, as a function of time and relative angular position. Activity is averaged across all reach & position neurons. Right: The normalized average firing rate in the 100 ms prior to target onset, plotted as a function of the relative angular hand position

reach up and right to get there (Fig. 6a). The monkeys performed a block of random-target reaches with normal feedback (baseline), followed by a block of movements with the VR. In this task, the statistics of reach directions relative to the hand position on the screen rotate by 30° (Fig. 6b). Thus, for the same cursor position in the workspace, the monkeys should plan movements 30° more clockwise than in the baseline condition. In fact, the monkeys did mostly learn to adapt their expectations of the upcoming movement directions, judging by their initial reach directions (Fig. 6c). We can therefore determine whether PMd activity changed to reflect the new movement probabilities at the same workspace positions.

Did PMd activity prior to target onset change to reflect the modified expected movement directions? Looking at individual neurons, the majority of PR neurons' preferred positions shifted counterclockwise in the VR task (Fig. 6d,e). This was expected, as the hand position most likely to result in a movement into a

neuron's PD will rotate counterclockwise in the VR task. There was a wide range of changes across neurons, again demonstrating that distributions are represented across the population of neurons.

To look at population level changes, we decoded reach direction using activity prior to target onset, and compared the distributions of these decoded reaches in the baseline and rotation periods of the task (Fig. 6f,g). As the rotation was small, we looked only at hand positions not near the center, where monkeys had more information about the upcoming movement directions. In every session for both monkeys, the decoded reach directions shifted clockwise relative to the workspace position. On average, there was a small, but significant, clockwise shift in decoding of about 10° for each monkey (Fig. 6d,e; $p < 0.05$ for both monkeys; for all decoded reaches see Supplementary Fig. 6a,b). Additionally, when we analyzed the data by looking at the average activity of PR neurons prior to target onset (as in Fig. 4a) rather than

decoding, we found a similar shift in activity in every experimental session (Supplementary Fig. 6c,d). Thus, PMd activity prior to target onset (see Supplementary Fig. 7 for effects after target onset) is modulated by the probabilities of upcoming movements, not simply by hand position for some other purpose.

**M1 does not represent a conditional probability distribution.** To determine whether primary motor cortex (M1) also represents conditional probability distributions of upcoming movements, we ran the same set of analyses for M1 as we did for PMd. When we did a GLM analysis, we found that 28% (176/618) of M1 neurons met the criteria of PR neurons, meaning they had significant modulation with hand position prior to target onset and movement after target onset. However, these neurons did not respond to hand position in the same manner as the PR neurons in PMd. These M1 neurons had increased activity prior to target onset when the hand position was in the same direction as the neurons' PDs, rather than in the opposite direction of the neurons' PDs (Fig. 7, Supplementary Fig. 8 for individual monkeys). This activity could be explained by the end of the previous movement, since previous movements into the PD (which correspond to higher M1 activity), often result in angular hand positions in the same direction as the PD (Supplementary Fig. 9). Moreover, the effect of the hand position didn't ramp up as it did in PMd; rather, it appears to be a decreasing effect from the previous movement (Fig. 7, Supplementary Fig. 8). Thus, while M1 activity varies according to position (as in refs. [12,13]), it likely does so in a way that reflects movement execution, rather than information about the upcoming movement.

**Discussion**
In order to plan everyday movements, we take into account the probability distributions of possible movements determined by the state of our body in the environment. Here, we have demonstrated that these conditional probability distributions influence behavior, specifically movement trajectories and latencies. A subpopulation of neurons in PMd, but not M1, function together to represent these probabilities, even prior to individual reaches. We used a visuomotor rotation task to show that the effect was not simply a position dependent component of the firing rate.

Probability distributions can be conditioned on many sources of information. Here, we focused on probability distributions that were determined by the body's state in the task environment. When hand position within the environment changed, the probability distributions of upcoming movements changed. Likewise, changes in the task environment caused by the visuomotor rotation altered the probability distributions. Although not the focus of our study, another source of probability distributions is biomechanics. In the extreme, biomechanics limit the possible movements. Softer constraints may arise from biomechanical costs, e.g., the ease of movement, which can affect both choice of arm movements[14,15] and PMd activity[15]. While biomechanics may influence the representation within PMd of possible upcoming movements, the fact that PMd's probability distributions also changed when the task environment changed in the VR task, suggests our findings are not solely due to biomechanical constraints.

In order to represent the probability distributions of upcoming movements, PMd needed to have information about the hand position in the workspace. Previous studies have shown that PMd activity is modulated by hand position, either to make the neurons' PDs compatible with the orientation of the arm[10], or to encode the relative position between the hand and eye[11]. Thus, there is evidence that PMd neurons have access to information

about hand position. In our task, PMd began to represent the possible upcoming movement while the current movement was ongoing and hand position was changing. Thus, if PMd was using proprioceptive information to determine the possible upcoming movements, it would have been using a changing position estimate. Alternatively, PMd could have made use of visual information about the current target, or an efference copy of the current movement command, to determine the probability distribution of the next movement. In everyday life, PMd likely uses a mixture of sensory, proprioceptive, and movement information to determine possible upcoming movements.

There is much debate on how probabilities are represented in the brain. Some argue for a temporal coding of uncertainty[16], while others argue that probability distributions are represented across populations of neurons[17–19] (e.g., probabilistic population coding[18]). Several previous studies have proposed models of movement in which distributed neural populations represent a probability density function across movement directions[20–24]. We showed that neurons were more strongly active in workspace locations from which movements into their preferred direction were more likely. Only when looking across neurons did the activity as a function of location closely match the task's movement probabilities. Also, when looking at neurons that were active at a nearby location, the distribution of PDs was proportional to the probability of upcoming movement directions from that location. Our results are consistent with the interpretation that coding of probability distributions across populations of neurons plays a central role in the movement decision process.

Still, our data are consistent with several possible interpretations of how the brain represents movement-related probabilities during planning. The fact that the decoded distributions are narrower when there are fewer possible upcoming movements could mean that PMd uses a continuous probability distribution to represent the potential movements at any given time. An equivalent, although semantically different, interpretation of the result is that PMd represents the uncertainty of the upcoming reach plan at any given time. A substantively different alternative is that the brain could be discretely sampling several of multiple possibilities from the probability distribution before a reach. For example, when there are few movement possibilities, the monkey might simultaneously preplan two movements, but when there are many possibilities, the monkey might preplan three or more movements. Another alternative interpretation is that the monkeys are rapidly sampling (and preplanning) individual reaches from the probability distribution at a rate much faster than 50 ms (the bin size used to calculate the distribution width in Fig. 5g). Future experiments with many more recorded neurons, resulting in more precise decoding, could help resolve these questions of how the probability distribution is represented.

When analyzing the visuomotor rotation task (Fig. 6), we found changes in PMd activity that corresponded to changes in the probability distribution of the upcoming movements, even though the visual distribution of targets did not change. On first glance, this would appear to contradict previous studies, which have suggested that PMd tracks the visual spatial parameters more than the actual movement direction[25–27]. However, we were analyzing PMd responses prior to target onset, while other studies have looked at PMd activity following target onset. In fact, when we analyzed PMd results after the target was displayed (Supplementary Fig. 7), we found that the visuomotor rotation did not change PMd's representation of the target, consistent with previous findings. It is only during the time prior to the target being visually displayed, that the activity of PR neurons in PMd changes to reflect the changing probabilities of the upcoming movements themselves.

Our study builds on much research about the role of PMd in planning upcoming movements[4,6,28–32]. Previous work has demonstrated that monkeys represent possible movements when selecting between alternatives[6,29–32] and when estimating the likely target location from visual cues[4]. These studies suggested that PMd can represent a probability distribution. Our work extends these findings by showing that PMd also represents dynamically changing probability distributions that are dependent on interactions between the body and the environment. Moreover, our work shows that PMd does not only represent probability distributions that are explicitly manipulated. Here, PMd represented probability distributions even in a standard reaching task that has been used in a variety of motor studies[33,34], where probabilities are usually considered to be irrelevant. The representation of conditional probability distributions of possible movements in PMd appears to be ubiquitous.

## Methods

**Behavioral paradigms.** Random-target experiment: Three monkeys (Monkeys M, T, and C) performed a random-target reaching task (similar to the experiments in refs. [33,34]) in which they controlled a computer cursor using arm movements (Fig. 1). Monkeys were seated in a primate chair while they operated a two-link planar manipulandum. Arm movements were constrained to a horizontal plane within a workspace of 20 cm × 20 cm. On each trial, the monkey consecutively reached to four targets (2 cm × 2 cm squares), with each new target appearing once the monkey reached the previous target. More precisely, once a target was reached, a new target was triggered 100 ms later, as long as the cursor remained on the target. The target appeared on-screen 96 ms after this trigger on average, due to delays from graphics processing and the monitor refresh rate. In accordance, the monkey was required to keep the cursor on the target for an additional 100 ms after a new target was triggered. Thus, in total there was a 200 ms hold period after landing on the target. This brief hold period forced the monkeys to decelerate as they approached the target, but was not so long that the monkeys completely stopped on the target. After a successful trial (four successful reaches), the monkey received a liquid reward. The next trial started after a delay of one second with a new random target presentation.

Target locations were chosen to be 5–15 cm from the current target. Specifically, they were chosen as follows. (1) Randomly choose a distance between 5 and 15 cm, and an angle between 0° and 360° for the new target (relative to the current target). (2) If the new target falls outside of the workspace, add 90° to the angle and set the distance to be 5 cm. (3) Repeat step 2 until the target is in the workspace.

Many of the analyses are aligned to target onset. These experiments did not use a photodiode to determine the exact moment the target was displayed. Thus, in all analyses, the target onset time we used was the time the computer sent the target command plus the average delay time (96 ms).

In total, we recorded eight sessions for monkey M, six sessions for monkey T, and five sessions for monkey C.

Visuomotor rotation experiment: Monkeys M and T each performed three sessions in which a visuomotor rotation (VR) task followed the baseline random-target task. The VR task was equivalent to the random-target task, with the exception that the movement vectors displayed on the screen (workspace) were rotated 30° counterclockwise relative to the hand movement vectors (as in Fig. 6a).

**Neural data acquisition and preprocessing.** Monkeys M and T were implanted with 100-electrode Utah arrays (Blackrock Microsystems, Salt Lake City, UT) in dorsal premotor cortex (PMd). Monkeys M and C were implanted with Utah arrays in primary motor cortex (M1). See ref.[4] for the location of the arrays in Monkeys M and T. Units were manually sorted with Offline Sorter (Plexon, Inc, Dallas, TX, USA). Only well-isolated individual units were included. Since we used chronically implanted arrays, it is likely that some neurons were recorded on multiple sessions and thus were not unique.

We only included neurons with firing rates of at least two spikes per second in either the early or late period. The early/late periods were defined as −100 to 50 and 50 to 200 ms from target onset, respectively. In PMd, this left us with 520 neurons from Monkey M and 250 neurons from monkey T. In M1, this left us with 352 neurons from Monkey M and 266 neurons from Monkey C.

**Animal care.** All procedures involving animals in this study were performed in accordance with the ethical standards of Northwestern University's Institutional Animal Care and Use Committee and are consistent with Federal guidelines.

**Behavioral analysis.** Each trial consisted of four reaches. We did not include the first reach in any of our analyses, as this was preceded by a reward period without movement (rather than being in the midst of a continuous movement). Reaches were also excluded if the monkey did not hold on the previous target for 200 ms, or

if it took greater than 1.4 s to reach the target. These "error" reaches were rare, and occurred 2.3%, 4.9%, and 0.8% of the time in monkeys M, T, and C, respectively. Behavioral data were combined across all sessions for each monkey.

Statistics of target presentation: We defined the angular position, $\phi_P$, as the hand position (prior to movement) relative to the center of the workspace (Fig. 1). We defined $\phi$ as the angular difference between the upcoming movement direction (also the direction to the target), $\phi_T$, and the angular position. That is, $\phi = \phi_T - \phi_P$.

Trajectory bias: We calculated whether the movement trajectory within a given time interval was biased toward the expected target direction, $\phi_E$. The expected direction is the most likely direction of the next target given the current hand position, based on the distribution of $\phi$'s. So if $\phi^*$ is the value corresponding to the circular mean of the distribution of $\phi$'s, $\phi_E = \phi_P + \phi^*$. The bias of the movement trajectory within a given time interval was defined as follows. First, a movement direction, $\phi_M$, was determined within that time interval based on the start and end hand position in that time interval. We calculated the bias, $B = \frac{\phi_M - \phi_T}{\phi_E - \phi_T}$, where the numerator and denominator were made to be in the interval of $[-180°\ 180°]$ prior to dividing. When the current movement direction is toward the expected direction, $B$ will be near 1, and when the movement direction is toward the actual target direction, $B$ will be near 0. $B$ can also be negative when the movement direction is away from the expected direction. For the summary statistics of $B$, we used the median and standard error of the median, as $B$ has outliers when dealing with circular variables. To calculate the standard error of the median, we used bootstrapping. Note that in Supplementary Fig. 1, we also calculated the bias toward the center, rather than the expected direction. This has the exact formulation as above, except $\phi^* = 180°$.

Latency effects: The latency of a reach was defined as the time from target onset until the movement surpassed a velocity of 8 cm s$^{-1}$. Latencies greater than 6 standard deviations from the mean were excluded as outliers.

We computed the mean latency of movements as a function of the expectedness of the target location, which was defined as the difference between the target direction and expected direction: $\left| \phi_T - \phi_E \right|$. We calculated the Pearson's correlation between latency and the expectedness of the movement, and determined significance based on the p-value associated with the correlation (two-sided one-sample t-test).

We also analyzed differences in latencies between expected reaches (expected direction < 60° from target direction) and unexpected reaches (expected direction > 120° from target direction), based on the distance of the hand position from the center of the workspace. To test whether the latency of expected reaches decreased as a function of distance from the center more than unexpected reaches, we used linear regression to fit the latency of reaches as a function of distance from the center. We then did a 2-sided unpaired t-test with unequal sample variances to analyze whether the slope was less (more negative) for expected reaches.

**Neural data analysis.** As with the behavioral analyses, we only included successful reaches, and did not include the first reach of each trial.

*Smoothed maps of neural activity*: For many aspects of the following neural data analysis, we computed smoothed maps of neural activity in relation to some variable (hand position, previous movement, or the upcoming movement). For instance, we created a map of how neural activity varied over all hand positions in the workspace, and a map of how neural activity varied in response to all upcoming movement vectors. For our maps, rather than assuming a parametric form, we non-parametrically estimated the average firing rate at each point in space using weighted k-nearest neighbor smoothing. The parameters were the number of nearest neighbors, k, and a decay parameter, d. As an example, for the movement variable (previous or upcoming), for each movement we found the k nearest movement vectors (based on Euclidean distance). We then averaged the firing rates associated with each of the k movements, but with each weighted proportional to its distance from the given movement vector to the d power.

The parameters we used for the generalized linear models (GLMs) were k = 20% of the data points, d = 0. The parameters were found using cross-validation on a held out data set, in order to not inflate the number of significant neurons in the GLM analysis. The parameters we used at other times (including in plots) were k = 30% of the data points, d = −1. These parameters were found using cross-validation on the current data sets in order to create as accurate maps as possible. Importantly, all results were robust to a wide range of smoothing parameters.

For visualizing the position maps in Fig. 4c, they were rotated either 90, 180, or 270° so that the PD of that neuron (after the same rotation) was always upward (between 45° and 135° relative to horizontal), which facilitated the interpretation and comparison of the results.

To get the estimated firing rate due to a single variable (e.g., position), we could use these smoothed maps. For any position we can get the associated estimated firing rate, $\theta_P$, by looking up the firing rate for that position on the smoothed map. If, for instance, we want to get the estimated firing rates due to position in a time interval prior to every reach, we would get a vector $\boldsymbol{\theta}_P$, which contains the estimate prior to each reach. The same can be done to estimate the firing rates due to the upcoming movements, $\boldsymbol{\theta}_{UM}$, or previous movements, $\boldsymbol{\theta}_{PM}$.

*Determining PDs of Neurons*: We determined the preferred movement direction (PD) of each neuron from 50 to 200 ms following target onset. Let $\boldsymbol{Y}$ be the vector of firing rates in that interval for every movement. It is possible that some of the

neural activity during these time periods was related to the hand position, rather than the upcoming movement. Thus, we first aimed to remove the effect of any position-related signal that might bias the calculated PD. Let $\theta_P$ be the vector of the estimated firing rates due to hand position in the same time interval (see Smoothed maps of neural activity section above for how we estimate $\theta_P$). We fit the tuning curves to $Y - \theta_P$, i.e., we subtracted out the position-related signal to get a "firing rate due to movement." More specifically, we fit a von Mises function to relate the movement directions to this "firing rate due to movement":

$$Y - \theta_P = \alpha \exp\left[\beta \cos\left(\phi_M - \phi_M^*\right)\right]$$

where $\phi_M$ is the vector of movement directions, and $\alpha$, $\beta$, and $\phi_M^*$ are the parameters that we fit. $\phi_M^*$ is the PD of the neuron.

*Preferred angular positions*: We determined the preferred angular (hand) position of each neuron from −100 to 50 ms following target onset. We first aimed to remove the effect of any movement-related signal that might bias the calculated preferred angular position. To do so, we subtracted the movement-related activity from the total activity (like in the previous section). We then fit a von Mises function to relate the angular positions to this residual. This fitting is identical to the fitting of PDs, except replacing movement directions with angular hand positions.

*PSTHs*: When plotting PSTHs of individual neurons, we plotted the mean firing rate across movements. The error bars on PSTHs are the standard error of the mean (SEM) across movements. When plotting the PSTHs averaged across neurons, we first normalized the mean firing rate (across time) for each neuron by dividing by the maximum firing rate of the average trace (across all conditions). We then plotted the average of these normalized firing rates across neurons. Error bars are the SEM across neurons. All traces were smoothed using a 50 ms sliding window.

PSTHs were made for different categories of movements. For the PSTHs, movements near the PD were those that were within 60° of the PD. Movements opposite the PD were those greater than 120° from the PD. Hand positions opposite the PD were angular positions greater than 120° away from the PD. Hand positions near the PD were angular positions less than 60° away from the PD.

*Generalized Linear Model*: To determine which variables were reflected in the neural activity, we used a Poisson Generalized Linear model (GLM). Let $Y$ be a vector containing the number of spikes in the time interval we are considering, for every movement. It has size $m \times 1$, where $m$ is the number of movements. We aimed to predict $Y$ based on several factors. We used the hand position, the previous movement vector, the upcoming movement vector, the peak velocity of the upcoming movement, and a baseline term. More specifically, the covariate matrix $X$ was:

$$X = \begin{bmatrix} | & | & | & | & | \\ 1 & \theta_P & \theta_{UM} & \theta_{PM} & v_{max} \\ | & | & | & | & | \end{bmatrix},$$

where $v_{max}$ is the vector of peak velocities of movements, and $\theta_P$, $\theta_{UM}$, and, $\theta_{PM}$ are generated from the smoothed maps (see Smoothed maps of neural activity above). Essentially, these covariates are the expected firing rates from position, upcoming movement, and previous movement (respectively) by themselves. Note that the previous and upcoming movement covariates were fit separately and do not need to have the same smoothed map (as PDs can be different during planning and movement[7–9]). Also note that when we run GLMs during different time intervals, we make separate smoothed maps for these time intervals.

Overall, the model that generates the firing rate ($\lambda$; also known as the conditional intensity function) can be written as:

$$\lambda = \exp(X\beta)$$

where $\beta$ is a vector of the weights for each covariate that we fit, and $X$ is the matrix of covariates, which is z-scored before fitting. If there are $j$ covariates, then $\beta$ has size $j \times 1$. $X$ has size $m \times j$. Note the use of an exponential nonlinearity to ensure that firing rates are positive. The model assumes that the number of spikes, $Y$, is generated from the firing rate, $\lambda$, according to a Poisson distribution.

We fit the model weights to the data using maximum likelihood estimation. That is, we found $\beta$ that was most likely to produce the true spike output (assuming spikes were generated from the firing rate in a Poisson nature). Critically, we used (5-fold) cross-validation, meaning that the model was fit to the data using one set of data (the training set), and model fits were tested with an independent set of data (the testing set). Similarly, when calculating the test set covariates for movement and position (described in Smoothed maps of neural activity), we only used k-nearest neighbors from the training set, to avoid overfitting.

To test whether an individual covariate significantly influenced neural activity, we first made sure that a simplified model with only that individual covariate had significant predictive power. To determine the value of a model fit, we used pseudo-$R^2$[35,36], a generalization of $R^2$ for non-Gaussian variables. The pseudo-$R^2$

of a model is defined as:

$$R_D^2(\text{model}) = 1 - \frac{\log L(n) - \log L\left(\hat{\lambda}\right)}{\log L(n) - \log L(\overline{n})}$$

where $\log L(n)$ is the log likelihood of the saturated model (i.e., one that perfectly predicts the number of spikes), $\log L\left(\hat{\lambda}\right)$ is the log likelihood of the model being evaluated, and $\log L(\overline{n})$ is the log likelihood of a model that uses only the average firing rate.

Then, in order to determine the importance of that covariate to the full model, we test whether the full model predicts neural activity significantly better than a model where that covariate is left out (reduced model). To compare the fits between the reduced model (model 1) and full model (model 2), we used relative pseudo-$R^2$, which is defined as:

$$R_D^2(\text{model 1, model 2}) = 1 - \frac{\log L(n) - \log L\left(\hat{\lambda}_2\right)}{\log L(n) - \log L\left(\hat{\lambda}_1\right)}$$

where $\log L\left(\hat{\lambda}_2\right)$ is the log likelihood of the full model and $\log L\left(\hat{\lambda}_1\right)$ is the log likelihood of the reduced model.

To determine significance, we bootstrapped the fits to create 95% confidence intervals, and checked whether the lower bounds of these confidence intervals were greater than 0. Note that the pseudo-$R^2$ and relative pseudo-$R^2$ values can be less than 0 due to overfitting.

*Neuron types*: In PMd, we defined selected-response (SR) neurons as those that were significantly modulated by upcoming movement in the late period in the GLM, but were not significantly modulated by hand position in the early period. Potential-response (PR) neurons were significantly modulated by upcoming movement in the late period and by hand position in the early period. Using a more relaxed criterion for defining neurons (as described in Decoding below) greatly increases the number of PR neurons. While our criteria for determining SR and PR neurons was different from ref. [6], we used the same terminology due to the same perceived function. In the VR task, PR neurons were those that were significant during both the baseline and VR periods.

Population activity over time averaged across trials: For each neuron, we calculated the firing rate as a function of the relative angular position (Fig. 4). We defined the relative angular position as the difference between a neuron's PD and the hand's angular position (the PD minus the angular hand position). In the VR task (Supplementary Fig. 6c), the relative angular position was calculated relative to the neuron's PD in the baseline task. We then normalized each neuron by dividing by its mean firing rate, and then averaged the normalized activity across neurons. We then smoothed the activity for plotting using the parameters from the smoothed maps.

We also made several variants of the above plot (Supplementary Fig. 5). We made a plot where movements were only used if the angle between previous and upcoming movements was less than 90°. We made a plot in which the movements were resampled, so that the distribution of $\phi$'s was centered at 180°, rather than being off center. More specifically, we resampled from a von Mises distribution: $g(\phi) \propto \exp[\cos(\phi - 180°)]$.

To determine the relative angular position resulting in peak activity (in the 100 ms before target onset), we calculated the activity at 20 relative angular positions (evenly spaced from 0 to 360°), and calculated the circular means of the angles weighted by their activities. We determined whether the activity prior to target onset was related to the distribution of upcoming movements by testing whether the relative angular position resulting in peak activity was significantly different from 150° (the circular mean of the distribution of $\phi$'s). We created a 95% confidence interval of peak relative angular positions by bootstrapping over the set of neurons, and checked whether this overlapped with 150°.

*Decoding*: We aimed to determine the movement intention of the neural population in the 100 ms prior to target onset. As only including PR neurons would give us a small number of neurons per session for decoding, we expanded our criteria. While PR neurons were significant for hand position and upcoming movement with 95% confidence, here we included neurons that were significant at a level of 50% (the median pseudo-$R^2$ and relative pseudo-$R^2$ values were greater than 0). For comparing between the VR task and baseline, we required that neurons had positive median pseudo-$R^2$ values in both conditions. Additionally, as neurons changed from session to session, separate decoders were trained for each session.

We first fit tuning curves to each neuron during 50 to 200 ms after target onset, when the neurons' preparatory responses to different target directions was known. This was done using a von Mises function, as in Determining PDs of neurons. We wanted to use these tuning curves to decode during the 100 ms prior to target onset. However, as firing rates were greatly different during these two time periods, we needed to rescale the tuning curves. To do so, we fit tuning curves to the future movement in the 100 ms prior to target onset. We then modified this tuning curve by giving it the preferred direction calculated after target onset. This essentially gives us rescaled versions of the tuning curves determined when the target is known.

Note that for decoding in the VR task, we still fit the initial tuning curve using activity after target onset in the baseline task. We then decoded using activity from before target onset in the VR task, using the rescaling described above. We use this procedure to make the comparison meaningful.

Next, for each neuron, we found the likelihood of the number of spikes given all possible movement directions (in 1° increments). This was done by assuming the number of spikes during the time period is a Poisson random variable with a mean rate determined by the value of the tuning curve at the direction being tested. If $r_i$ is the number of spikes during the interval for neuron $i$, $s$ is the direction, and $f_i(s)$ is the value of the tuning curve (the expected number of spikes) for neuron $i$ at direction $s$:

$$P(r_i|s) = \frac{\exp[-f_i(s)]f_i(s)^{r_i}}{r_i!}$$

We assumed that neurons' activities were conditionally independent given the direction (a naïve Bayes decoder), and thus multiplied their probability distributions:

$$P(\boldsymbol{r}|s) \propto \prod_i P(r_i|s)$$

We can use Bayes rule to determine the likelihood of all the movement directions given the number of spikes of all neurons. Assuming a uniform prior, by Bayes rule:

$$P(s|\boldsymbol{r}) \propto P(\boldsymbol{r}|s)$$

Finally, we normalized $P(s|\boldsymbol{r})$ (so it was a probability distribution), and this term was the decoded distribution.

The decoded direction was the direction corresponding to the peak of the distribution (the maximum likelihood decoded direction). The width of the decoded distribution was the full width half maximum (FWHM) of the decoded distribution.

To calculate the width of the decoded distribution over time (Fig. 5g), we decoded using a 50 ms sliding window of neural activity. All methods were the same as above, just replacing the 100 ms of activity prior to target onset with the given 50 ms of activity. This choice allowed us a better temporal resolution.

As we did decoding separately for each session, to do significance testing, we used a simple multilevel model analysis—specifically, a random intercepts model, where the baseline (intercept) can be different for every session. Thus, if there were 4 sessions, when looking at the width of the decoded distribution ($w$) as a function of the distance from center ($d$), we wrote the model as $w = \beta_1 I_1 + \beta_2 I_2 + \beta_3 I_3 + \beta_4 I_4 + \beta_5 d$, where $I_1$ through $I_4$ are indicator variables for whether the values are from a given session. We looked at whether $\beta_5$ was significantly different from 0 using a two-sided one-sample $t$-test. We used an equivalent approach to determine the significance of the relationship between latency and the difference between the decoded direction and the target direction. We also used an equivalent approach to determine the significance of the relationship between latency and the width of the distribution for expected reaches (expected reaches are those where the difference between the actual and expected target direction is less than 60°).

We calculated the bias of the initial trajectory toward the decoded direction, $\phi_D$, equivalently to how we calculated the behavioral bias toward the expected target direction: $B = \frac{\phi_M - \phi_T}{\phi_D - \phi_T}$.

To determine whether the distribution of decoded reach directions shifted from the baseline task to the VR task, we calculated the difference between the circular means of the distributions of "decoded $\Phi$'s" (the decoded reach direction relative to the angular position). For significance testing, we bootstrapped this difference in circular means. More specifically, 1000 times, we resampled decoded reaches within the baseline task and calculated the baseline circular mean, and did the same thing for the VR task. This led to 1000 differences in circular means. We looked at the 95% confidence interval of this difference.

**Code availability**. All code is available from the authors upon request.

**Data availability**. All data are available from the authors upon request.

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

## Acknowledgements

We would like to thank Brian Dekleva, Daniel Wood, Pat Lawlor, Mark Segraves, Hugo Fernandes, and Xuelong Zhao for helpful comments and discussions. For funding, J.I.G. would like to thank NIH F31 EY025532 and NIH T32 HD057845. M.G.P. would like to thank NIH F31 NS092356 and NIH T32 HD07418. P.R., L.E.M., and K.P.K. would like to thank NIH R01 NS074044.

## Author Contributions

J.I.G., M.G.P., L.E.M., and K.P.K. designed the study. M.G.P. ran the experiments. J.I.G., M.G.P., and P.R. analyzed the results. J.I.G. and K.P.K. wrote the manuscript. All authors edited and reviewed the manuscript.

## Additional information

**Competing interests:** The authors declare no competing interests.

