## [Peer Review File · Nature Communications]

Editorial Note: This manuscript has been previously reviewed at another journal that is not operating a transparent peer review scheme. This document only contains reviewer comments and rebuttal letters for versions considered at Nature Communications. Reviewer #4 was added for the review process at Nature Communications.

Reviewers' comments:

Reviewer #3 (Remarks to the Author):

This paper evaluates how neurons in two areas of the motor system, PMd and M1, change their firing behavior as a function of hand position and upcoming movement direction. The authors argue that neurons in PMd, but not M1, encode the conditional probability distribution of upcoming movements.

I reviewed an earlier version of this work, and the authors have addressed all of my previous comments. Overall, this paper is written at a high level and the premise is very compelling. This paper gets at the core question of motor neurophysiology: how is neural activity related to motor planning and execution? In the motor system, neurons have been found to encode a multitude of movement parameters: velocity, position, force, speed, etc. This paper helps us reevaluate those findings, providing a coherent explanation of how those variables might be related, at least for the case of velocity and position. The analyses suggesting that a subset of PMd neurons encodes conditional movement probability is cleverly done, with appropriate statistical controls. I believe this paper will have a substantial impact on the field.

I have only one relatively minor comment outstanding. With respect to the null result of neurons in M1 *not* representing conditional probability distributions: how many neurons in M1 were evaluated? Given the relatively rare instance of unambiguous PR response in PMd, how many neurons in M1 would have been expected to show the effect? Some kind of statistical power analysis that suggests that if the effect were there it would have been seen would be appropriate.

Reviewer #4 (Remarks to the Author):

In this manuscript, the authors suggest that neural activity in PMd prior to target onset reflects the monkey's estimate of the conditional probability distribution of potential reach targets. This is a reasonable proposal, but I was skeptical that it can be convincingly demonstrated, especially in a study using random targets presented in a limited workspace. In short, I had many of the same misgivings expressed by the previous reviewers regarding potential confounds between the distribution of target directions and positional "gain fields", biomechanical constraints, across-trial averaging effects, etc. Nevertheless, the analyses presented here convinced me, one-by-one, that these alternative interpretations can indeed be rejected, leaving strong evidence in favor of the authors' hypothesis.

In my opinion, three aspects of the results make them most convincing: 1) The distribution of directions decoded from neural activity before target onset matched very well the probability distribution of actual target directions, notably including a clockwise bias that was entirely due to task design; 2) the activity changed in the predicted way when cursor and hand movement were dissociated with a 30 degree visuomotor rotation; 3) these effects were present only in PMd and entirely absent in M1, which would be expected to be more subject to confounds such as biomechanics.

In short, I think this is beautiful data that leads to a very important conclusion that warrants publication. I only have a few minor suggestions for further improvements and clarifications:

Minor comments:

I agree with the authors that the 156 degree bias is a very important feature of the task design, and I would encourage them to emphasize even more strongly the fact that this bias is seen in the neural activity (Fig. 4a, 5b). Seeing that is what I found most convincing.

The authors show a very compelling match between the distribution of phi angles across all movements (Figure 4b) and the distribution of angular position relative to PD (Figure 4a). Again, most convincing is that both of these exhibit a bias to 156 degrees, which is a consequence of the algorithm that picked targets and not of other factors such as biomechanics or a tendency toward the center. Furthermore, the decoded angles showed the same tendency (Fig 5b), but not when examined for positions near the center (Fig 5c) – as expected given Fig 1g. Indeed, that distribution from central space in Fig 5c looks remarkably like the distribution in Fig 1g, with a strongest mode to quadrant 2, smaller modes to quadrants 1 and 3, and the smallest to quadrant 4. That almost looks too good to be true, and invites all kinds of skepticism – but because this decoding is done on the basis of activity *before* target onset, it can't be an artifact of actual target distribution except insofar as PMd is sensitive to that distribution. In other words, it really supports the authors' hypothesis. I'd invite the authors to test whether that match is statistically significant and if so, emphasize it in the text.

Perhaps even more impressive is the good match between the decoded directions from different regions of space (Fig. 5e). However, I don't understand the statement: "These distributions are constructed to have the average width and peak angle (according to circular statistics) of all reaches from the hand positions within the grid square". If they are constructed to match the reach direction distributions then their match to reach direction distributions doesn't mean anything. I must not understand what you did here. I would have thought the distributions would be driven by the neural decoding data, as in Fig 5b and c. Or are you saying that you picked trials from recording sessions such that the distribution of actual target locations from those trials matched the distribution from all trials observed in all sessions, including training – and only then derived neural activity distributions on the basis of activity prior to target onset? If the previous sentence is confusing, that's because I'm confused, so please clarify.

For completeness, it would be useful to show the data for reaches made near the center in the same format as Fig. 4 a&b, perhaps just below those panels.

Figure 2: I'd add the behavioral data for monkey C here (in the same format for panels c & d). It's not as beautiful as the others, but still convincing and supports reproducibility. I would also suggest "labeling" the data sets here as "Monkey M (PMd and M1)", "Monkey T (PMd)", and "Monkey C (M1)". That way readers will not be confused, as I was, why we're not seeing the data from all three monkeys when you look at neural data. It's in the methods, I know, but by then it's too late.

One thing I found disturbing is that the activity at target onset (e.g. dashed line in Fig. 3abc) already differed when the target appeared in the cell's PD versus the opposite direction. But how could it if the monkey did not yet see the target? Were the 50ms sliding windows overlapping (I assume not)? Might this be due to different success rates in reaching toward expected versus unexpected targets? Or could it be due to the inaccuracy of target onset timing – i.e. sometimes the target appeared less than 96ms after the command, which was your estimate of the lag in the display?

Somewhere it should be reported how often the monkeys made reaching errors as a function of how close the target direction was to the expected direction. I expect that might show trends congruent with the RT data, but even if not, it would be useful for the sake of completeness.

Can the authors comment on why the distribution of normalized firing rates prior to target onset is rotated CCW away from 0 degrees in M1? I assume the "rotation" after target onset is just due to the weak relations between pre- versus post-onset PDs.

Reviewer 1 raised an interesting point about previous studies showing cursor, not hand, direction most strongly represented in PMd. The authors' response is reasonable, but I think it would be useful to include it in the main text discussion instead of just in the supplemental data.

In the discussion, the authors should mention models by Steven Tipper (1998 Phil Trans), Erlhagen & Schoner (2002 Psych Rev), Cisek (2006 J Neurosci), Furman & Wang (2008 Neuron) and Klaes, Schneegans, Schoner & Gail (2012 Plos). While those models did not explicitly predict the present result, they all suggested the idea of distributed neural populations representing a probability density function across directions. The data shown here can be taken as strong evidence in favor of that proposal.

Page 10, first column: It would be useful to add "cursor" before "position in the workspace".

Page 10, second column: "looked" -> "looking"

Figure 7 caption: "not evidence" -> "no evidence"

We would like to thank both reviewers for their helpful comments. We have responded to all the comments individually below. Moreover, following the reviewers' comments, we list additional minor changes we have made to the manuscript for full transparency.

Reviewers' comments:

Reviewer #3 (Remarks to the Author):

This paper evaluates how neurons in two areas of the motor system, PMd and M1, change their firing behavior as a function of hand position and upcoming movement direction. The authors argue that neurons in PMd, but not M1, encode the conditional probability distribution of upcoming movements.

I reviewed an earlier version of this work, and the authors have addressed all of my previous comments. Overall, this paper is written at a high level and the premise is very compelling. This paper gets at the core question of motor neurophysiology: how is neural activity related to motor planning and execution? In the motor system, neurons have been found to encode a multitude of movement parameters: velocity, position, force, speed, etc. This paper helps us reevaluate those findings, providing a coherent explanation of how those variables might be related, at least for the case of velocity and position. The analyses suggesting that a subset of PMd neurons encodes conditional movement probability is cleverly done, with appropriate statistical controls. I believe this paper will have a substantial impact on the field.

I have only one relatively minor comment outstanding. With respect to the null result of neurons in M1 *not* representing conditional probability distributions: how many neurons in M1 were evaluated? Given the relatively rare instance of unambiguous PR response in PMd, how many neurons in M1 would have been expected to show the effect? Some kind of statistical power analysis that suggests that if the effect were there it would have been seen would be appropriate.

In PMd, 99/770 (13%) of neurons were potential-response neurons. That is, their activities were significantly modulated by the upcoming movement after target onset and by hand position prior to target onset. Interestingly, in M1, even more neurons (176/618 = 28%) met these same significance criteria. In other words, more neurons in M1 than PMd were modulated by position prior to target onset, so the reason we had a null M1 result was not due to limited M1 neurons. The reason was because these M1 neurons did not represent position in the same manner as do those of PMd. As shown in Fig. 7 (PSTH row 2 and panel E), the activity of these M1 neurons is not greater when the hand is in positions that increase the probability of an upcoming movement into their PD. Rather, it appears that these M1 neurons execute movement rather than provide information about an upcoming movement.

We have added this information to the main text and clarified in the figure caption that the “reach & position” neurons of M1 met the same significance criteria as PMd's potential-response neurons.

Reviewer #4 (Remarks to the Author):

In this manuscript, the authors suggest that neural activity in PMd prior to target onset reflects the monkey's estimate of the conditional probability distribution of potential reach targets. This is a reasonable proposal, but I was skeptical that it can be convincingly demonstrated, especially in a study using random targets presented in a limited workspace. In short, I had many of the same misgivings expressed by the previous reviewers regarding potential confounds between the distribution of target directions and positional "gain fields", biomechanical constraints, across-trial averaging effects, etc. Nevertheless, the analyses presented here convinced me, one-by-one, that these alternative interpretations can indeed be rejected, leaving strong evidence in favor of the authors' hypothesis.

In my opinion, three aspects of the results make them most convincing: 1) The distribution of directions decoded from neural activity before target onset matched very well the probability distribution of actual target directions, notably including a clockwise bias that was entirely due to task design; 2) the activity changed in the predicted way when cursor and hand movement were dissociated with a 30 degree visuomotor rotation; 3) these effects were present only in PMd and entirely absent in M1, which would be expected to be more subject to confounds such as biomechanics.

In short, I think this is beautiful data that leads to a very important conclusion that warrants publication. I only have a few minor suggestions for further improvements and clarifications:

Minor comments:

I agree with the authors that the 156 degree bias is a very important feature of the task design, and I would encourage them to emphasize even more strongly the fact that this bias is seen in the neural activity (Fig. 4a, 5b). Seeing that is what I found most convincing.

When describing Fig. 4a and 5b, we modified the text slightly to emphasize this more.

The authors show a very compelling match between the distribution of phi angles across all movements (Figure 4b) and the distribution of angular position relative to PD (Figure 4a). Again, most convincing is that both of these exhibit a bias to 156 degrees, which is a consequence of the algorithm that picked targets and not of other factors such as biomechanics or a tendency toward the center. Furthermore, the decoded angles showed the same tendency (Fig 5b), but not when examined for positions near the center (Fig 5c) – as expected given Fig 1g. Indeed, that distribution from central space in Fig 5c looks remarkably like the distribution in Fig 1g, with a strongest mode to quadrant 2, smaller modes to quadrants 1 and 3, and the smallest to quadrant 4. That almost looks too good to be true, and invites all kinds of skepticism – but because this decoding is done on the basis of activity *before* target onset, it can't be an artifact of actual target distribution except insofar as PMd is sensitive to that distribution. In other words, it really supports the authors' hypothesis. I'd invite the authors to test whether that match is statistically significant and if so, emphasize it in the text.

We agree that the general match between the behavioral and neural distributions is very interesting. However, we do need to be cautious in claiming that the precise details of the distributions of the same.

First, we have replotted the behavioral distributions in Fig. 1 so that they have the same bin size (10°) as the neural distributions. Looking at the updated figures, while it is clear that the behavioral and neural distributions have the same overall structure, we feel it is overambitious to claim that all the details are the same.

Moreover, we did a Kuiper's test (circular statistics equivalent of Kolmogorov-Smirnov test) to compare the behavioral distributions in Fig. 1f,g with those in Fig. 5b,c. Even though both Fig. 1f is similar to Fig. 5b, and Fig. 1g is similar to Fig. 5c, we got different statistical results. When using all reaches (Figs. 1f and 5b), a significant difference was found between distributions. When only using reaches starting near the center (Figs. 1g and 5c), no significant difference was found. However, the primary reason for this difference in significance results is the differing sample sizes. When using all reaches, we had a sample size $> 11,000$, so that a small difference in the distributions could be shown to be significant. When using only reaches starting near the center, we had a sample size < 600 , in which case a small difference in distributions could not be shown to be significantly different. Given these nuanced results of statistical significance testing, which could be misleading to readers, we think it is better to not include them in the manuscript. In the main text, we continue to claim that the behavioral and neural distributions are similar, but we do not claim a near-exact match.

Perhaps even more impressive is the good match between the decoded directions from different regions of space (Fig. 5e). However, I don't understand the statement: "These distributions are constructed to have the average width and peak angle (according to circular statistics) of all reaches from the hand positions within the grid square". If they are constructed to match the reach direction distributions then their match to reach direction distributions doesn't mean anything. I must not understand what you did here. I would have thought the distributions would be driven by the neural decoding data, as in Fig 5b and c. Or are you saying that you picked trials from recording sessions such that the distribution of actual target locations from those trials matched the distribution from all trials observed in all sessions, including training - and only then derived neural activity distributions on the basis of activity prior to target onset? If the previous sentence is confusing, that's because I'm confused, so please clarify.

We apologize for the confusion. The distributions in Fig. 5e are driven by the neural decoding data, as in all other panels in Fig. 5. Within each grid square of the plot, the width of the displayed distribution is the average width of all single-trial decoded distributions. Likewise, the peak angle of the displayed distribution is the average peak angle of all single-trial decoded distributions. We have changed the wording accordingly.

For completeness, it would be useful to show the data for reaches made near the center in the same format as Fig. 4 a&b, perhaps just below those panels.

Thanks for the suggestion. We have added these panels, so Fig. 4 c&d now show reaches made near the center in the same format as Fig. 4 a&b, and we have added corresponding text. We have

also included these plots for the individual monkeys as the bottom row in Supplementary Fig. 5 rather than the heat map using residuals (what was previously in the bottom row).

Figure 2: I'd add the behavioral data for monkey C here (in the same format for panels c & d). It's not as beautiful as the others, but still convincing and supports reproducibility. I would also suggest "labeling" the data sets here as "Monkey M (PMd and M1)", "Monkey T (PMd)", and "Monkey C (M1)". That way readers will not be confused, as I was, why we're not seeing the data from all three monkeys when you look at neural data. It's in the methods, I know, but by then it's too late.

We have followed these suggestions, and have also mentioned the brain areas associated with each monkey at the beginning of the results section.

One thing I found disturbing is that the activity at target onset (e.g. dashed line in Fig. 3abc) already differed when the target appeared in the cell's PD versus the opposite direction. But how could it if the monkey did not yet see the target? Were the 50ms sliding windows overlapping (I assume not)? Might this be due to different success rates in reaching toward expected versus unexpected targets? Or could it be due to the inaccuracy of target onset timing - i.e. sometimes the target appeared less than 96ms after the command, which was your estimate of the lag in the display?

For potential response (PR) neurons, the reason for the neural difference before target onset between reaches towards and away from the PD is because the monkey was able to anticipate the direction of the upcoming reach because of its strong dependence on current hand position. We have clarified this point in the main text.

For selected response (SR) neurons there are two different reasons for the slight differentiation of neural activity prior to target onset. We now show and describe this in Supplementary Figure 3, copied below:

Supplementary Figure 3: Explaining pre-target activity for SR neurons

In Fig. 3, for SR neurons, prior to target onset there began to be a slight separation between activity traces depending on whether the reach will be near vs. opposite the PD. Given that SR neurons are supposed to only respond after target onset, this is initially surprising. However, there are two likely reasons for SR neurons' apparent pre-target activity. The first reason is our classification criteria of SR and PR neurons. PR neurons, unlike SR neurons, were significantly modulated by hand position prior to target onset (see *Methods* for details). That is, if a neuron was modulated by hand position with > 95% (e.g. 96%) confidence, then it was a PR neuron, but if it was modulated by hand position with < 95% (e.g. 94%) confidence, it would be an SR neuron. Thus, using this "conservative classification" (as we do in Fig. 3), we are likely including some PR neurons in the SR category. A PSTH using this conservative classification, copied from Fig. 3, is shown in the top row. Instead, if we use a "relaxed classification" that includes neurons as PR neurons if they are modulated by hand position with > 50% confidence, then SR neurons should not include any true PR neurons. When we plot SR neurons using this relaxed classification (bottom row), the differential activity prior to target onset diminishes, demonstrating that some PR neurons being included as SR neurons was a cause of the differential activity. Note that Supplementary Fig. 4 gives more details about different "conservative" and "relaxed" classification types. A second reason for the pre-target-onset differentiation of SR neurons in Fig. 3 is jitter in the time of target onset. While we subtracted the average delay for the target to be displayed on screen, there was some jitter in this timing (see *Methods*). Thus, some activity aligned to target onset could appear slightly earlier than it occurred.

Somewhere it should be reported how often the monkeys made reaching errors as a function of how close the target direction was to the expected direction. I expect that might show trends congruent with the RT data, but even if not, it would be useful for the sake of completeness.

Our task was designed to be easy for the monkeys, as they simply needed to reach to a single, clearly presented target. Thus, there were no errors in which a monkey reached to the wrong target.

However, we did exclude reaches in two circumstances. First, we excluded reaches if the monkey did not hold on the previous target for 200 ms, which meant that they were not following task

instructions. Second, we excluded reaches if they took greater than 1.4 seconds to reach the target, which generally happened when the monkey wasn't paying attention to the task. These two types of errors were infrequent, and occurred 2.3%, 4.9%, and 0.8% of the time in monkeys M, T, and C, respectively. We have now included a more thorough description of these types of errors leading to exclusion in the Methods / Behavioral Analysis section.

Can the authors comment on why the distribution of normalized firing rates prior to target onset is rotated CCW away from 0 degrees in M1? I assume the "rotation" after target onset is just due to the weak relations between pre- versus post-onset PDs.

A portion of the CCW bias can be explained by the correlation between the previous movement and resulting hand position. We have included a new supplementary figure (Supp. Fig. 9) that shows a similar bias in the distribution of angles between the previous movement and resulting hand position. That being said, this behavioral correlation does not explain all of the ~45° CCW bias in the M1 activity. The distribution of these angles between previous movement and the resulting hand position is peaked at ~10°, with a circular mean of 23°. The source of the remaining CCW bias is unclear, though it may have to do with biomechanics.

Reviewer 1 raised an interesting point about previous studies showing cursor, not hand, direction most strongly represented in PMd. The authors' response is reasonable, but I think it would be useful to include it in the main text discussion instead of just in the supplemental data.

We have now added a section to the main text discussion, titled "Dissociation between visual and motor responses in the visuomotor rotation task."

In the discussion, the authors should mention models by Steven Tipper (1998 Phil Trans), Erlhagen & Schoner (2002 Psych Rev), Cisek (2006 J Neurosci), Furman & Wang (2008 Neuron) and Klaes, Schneegans, Schoner & Gail (2012 Plos). While those models did not explicitly predict the present result, they all suggested the idea of distributed neural populations representing a probability density function across directions. The data shown here can be taken as strong evidence in favor of that proposal.

Thank you for all of the relevant citations, and the excellent summary of these studies. We have now mentioned these studies in the discussion subsection "Representation of probability distributions in the brain".

Page 10, first column: It would be useful to add "cursor" before "position in the workspace".

We have made this change.

Page 10, second column: "looked" -> "looking"

We have corrected this.

Figure 7 caption: "not evidence" -> "no evidence"

We have corrected this.

Additional minor changes:

Here are the additional minor changes we have made to the manuscript resulting from 1) comments from other readers on the manuscript preprint and 2) double-checking all analysis code. There were no qualitative changes or changes to the significance of results due to these updates.

- 1) The circular mean of the distribution of Φ 's (the angle between the upcoming movement and position vectors) is 150° . Rather than the circular mean, we previously listed the linear mean (156°). We now use 150° , to be consistent with other analyses. Neural activity as a function of the relative position angle (Fig. 4a) is actually centered at 151° , closer to the corrected statistic. This change has also led to minor differences in our behavioral results (Fig. 2 and statistics in main text) and Fig. 6b.
- 2) We previously incorrectly reported that we split the distances from the center into 4 quartiles for plotting in Fig. 2b,d and Fig. 4f,g. They are actually defined as follows (which is corrected in the caption of Fig. 2): "closest" is 0-20% of distances from the center, "mid-close" is 20-40%, "mid-far" is 40-60%, and "farthest" is 60-100%. The problem with using standard quartiles is as follows. In Fig. 2d, we plot the difference in mean latency between "expected" and "unexpected" reaches, depending on the hand's distance from the center. However, when using standard quartiles, there were no "unexpected" reaches for some monkeys in the last quartile (greatest 25% of distances from the center). This is because when very far from the center, the next target cannot be in an unexpected direction (farther away from the center). Thus, our binning ensures that each bin has unexpected reaches that can be plotted.
- 3) We have updated the example neurons in Fig. 3. Previously, for these example neurons, we had arbitrarily chosen reaches and hand positions that were near the PD and opposite the PD. Now we have chosen them automatically using the same criteria (described in the Methods) used when creating PSTHs averaged across neurons. Moreover, we have given error bars to the GLM results for these example neurons.
- 4) In Supplementary Fig. 5, we have replotted rows 2 and 3 due to a minor error in our resampling methodology.
- 5) For Fig. 4e, we rotate the position maps so that the PD is oriented upwards (as described in Methods). Before, when position activity maps were supposed to be rotated 180 degrees, we flipped them vertically instead. We have corrected this, and replotted the panel. Note that the main difference that can be observed in the panel is simply a different randomization of the placement of the example neurons.
- 6) In the Methods, we wrote that we trained the decoder using data from 50 to 200 ms after target onset. However, for some results, we had trained the decoder from 50 to 150 ms. We have corrected this so that the decoder was always trained from 50 to 200 ms after target onset.
- 7) We have referenced how our behavioral trajectory findings relate to the behavioral movement literature.
- 8) We have mentioned how our behavioral trajectory and latency results are not independent of each other.
- 9) We have been more specific in the Methods about the statistical tests we performed (e.g. writing "two-sided unpaired t-test with unequal sample variances" rather than "two-sided t-test").

REVIEWERS' COMMENTS:

Reviewer #4 (Remarks to the Author):

The authors have addressed all of my previous comments and I have nothing more to add. In my opinion this is a very good paper that will make a significant contribution to the field.